# Efficient Viral Transduction in Fetal and Adult Human Inner Ear Explants with AAV9-PHP.B Vectors

**DOI:** 10.3390/biom12060816

**Published:** 2022-06-10

**Authors:** Edward S. A. van Beelen, Wouter H. van der Valk, Thijs O. Verhagen, John C. M. J. de Groot, Margot A. Madison, Wijs Shadmanfar, Erik F. Hensen, Jeroen C. Jansen, Peter Paul G. van Benthem, Jeffrey R. Holt, Heiko Locher

**Affiliations:** 1Department of Otorhinolaryngology and Head & Neck Surgery, Leiden University Medical Center, Albinusdreef 2, 2333 ZA Leiden, The Netherlands; e.s.a.van_beelen@lumc.nl (E.S.A.v.B.); w.h.van_der_valk@lumc.nl (W.H.v.d.V.); t.o.verhagen@lumc.nl (T.O.V.); j.c.m.de_groot@lumc.nl (J.C.M.J.d.G.); e.f.hensen@lumc.nl (E.F.H.); j.c.jansen@lumc.nl (J.C.J.); p.van.benthem@lumc.nl (P.P.G.v.B.); 2Department of Otolaryngology & Neurology, Boston Children’s Hospital and Harvard Medical School, Boston, MA 02115, USA; margot.madison@childrens.harvard.edu; 3Mildred Clinics, 5611 PK Eindhoven, The Netherlands; wijss@mildredclinics.nl; 4The Novo Nordisk Foundation Center for Stem Cell Medicine (reNEW), Leiden University Medical Center, 2333 ZA Leiden, The Netherlands

**Keywords:** inner ear, hair cell, cochlea, vestibular, utricle, saccule, ampulla, hearing loss, gene therapy, AAV, AAV9-PHP.B

## Abstract

Numerous studies have shown the recovery of auditory function in mouse models of genetic hearing loss following AAV gene therapy, yet translation to the clinic has not yet been demonstrated. One limitation has been the lack of human inner ear cell lines or tissues for validating viral gene therapies. Cultured human inner ear tissue could help confirm viral tropism and efficacy for driving exogenous gene expression in targeted cell types, establish promoter efficacy and perhaps selectivity for targeted cells, confirm the expression of therapeutic constructs and the subcellular localization of therapeutic proteins, and address the potential cellular toxicity of vectors or exogenous constructs. To begin to address these questions, we developed an explant culture method using native human inner ear tissue excised at either fetal or adult stages. Inner ear sensory epithelia were cultured for four days and exposed to vectors encoding enhanced green fluorescent protein (eGFP). We focused on the synthetic AAV9-PHP.B capsid, which has been demonstrated to be efficient for driving eGFP expression in the sensory hair cells of mouse and non-human primate inner ears. We report that AAV9-PHP.B also drives eGFP expression in fetal cochlear hair cells and in fetal and adult vestibular hair cells in explants of human inner ear sensory epithelia, which suggests that both the experimental paradigm and the viral capsid may be valuable for translation to clinical application.

## 1. Introduction

The worldwide prevalence of sensorineural hearing loss (SNHL) is greater than that of all neurological disorders combined [1]. SNHL is caused by genetic or acquired dysfunction of the inner ear, the eighth cranial nerve, or both. Congenital hearing loss occurs in at least 1:1000 newborns, half of which can be attributed to inherited genetic changes [2]. The current standard of care involves the amplification of sound by use of hearing aids or cochlear implants via electrical stimulation of the cochlear nerve, but this only partially restores auditory function and only for a subset of patients [3]. In addition, disorders of the inner ear may include the vestibular system as well. The vestibular organs contribute to the sense of balance, visual gaze stability, and posture. Loss of vestibular function evokes imbalance and vertigo. Despite major advancements in the nascent field of inner ear gene therapy, few therapeutic strategies have been tested in human inner ear tissue [4,5,6]. 

One particularly promising approach to prevent hearing loss or restore auditory function is the use of DNA constructs delivered by viral vectors. A growing body of evidence supports inner ear gene therapy as a novel treatment option for SNHL, as recently reviewed elsewhere [7,8,9,10,11,12,13,14,15,16,17]. However, clinical translation is significantly challenged by: (1) a lack of human in vitro or ex vivo data, and (2) the inability of conventional adeno-associated viral (AAV) vectors to efficiently transduce the specific target cell(s). To address these issues, we tested the transduction efficacy of a synthetic AAV capsid encoding enhanced green fluorescent protein (eGFP) in human fetal and adult inner ear explants using a novel culture method. We found that our AAV2/AAV9-PHP.B-CMV-eGFP-WPRE construct can efficiently transduce cells in the fetal and adult human inner ear, including sensory hair cells. Our data suggest that AAV9-PHP.B vectors might be promising candidates for targeted gene delivery for the treatment of genetic and acquired inner ear disorders in patients. Furthermore, we suggest that our human inner ear culture method may help validate gene therapy reagents optimized for other model systems. 

## 2. Materials and Methods

### 2.1. AAV9-PHP.B Vectors 

AAV2/AAV9-PHP.B-CMV-eGFP-WPRE vectors were generated as previously described [14]. In short, vectors were generated and produced by the Viral Core at Boston Children’s Hospital (BCH) in compliance with the BCH Institutional Biosafety Committee (protocol # IBC-P00000447). AAV vectors were purified by iodixanol gradient ultracentrifuge, followed by ion exchange chromatography. Titers were calculated by quantitative PCR with eGFP primers (F-GACCTTTGGTCGCCCGGCCT, R-GAGTTGGCCACTCCCTCTCTGC). The titer of AAV2/AAV9-PHP.B-CMV-eGFP-WPRE was 3.5 × 10^13^ gc/mL. Vectors were aliquoted and stored at −80 °C.

### 2.2. Inner Ear Sample Collection, Dissection, and Viral Transduction

Human adult vestibular samples were collected from 5 patients (ages ranging from 33 to 62; 4 female, 1 male) undergoing translabyrinthine surgery for vestibular schwannomas. Air cells in the mastoid were drilled out to reach the bony labyrinth (Figure 1). The lateral and superior semicircular canals were identified, and their ampullas and subsequently the macula of the utricle were carefully excised. Samples were immediately transported in DMEM/F-12 supplemented with 0.5× N2, 0.5× B-27 (minus vitamin A) and Normocin (100 µg/mL). Using a Leica M205C dissecting microscope, debris and the utricular otolithic membrane with the attached otoconia were removed (Figure 2A,B). Two complete adult ampullas and three complete adult utricles were collected. The other samples were excluded from further experiments due to damage caused during excision from the otic capsule. 

Human fetal inner ear samples of 14 weeks fetal age were collected after elective termination of pregnancy by vacuum aspiration. Fetal age (in weeks, W), defined as the duration since fertilization, was determined by obstetric ultrasonography prior to termination, with a standard error of two days. Samples were transported in DMEM/F-12, supplemented with 0.5× N2, 0.5× B-27 (minus vitamin A) and Normocin (100 µg/mL), and dissected directly upon arrival in the laboratory using a Leica M205C dissecting microscope. After removing the bony capsule and identifying the otolith organs, ampullas, and the cochlear duct, the endolymphatic compartment was opened to expose the sensory domains (Figure 2C–E). A total of three complete fetal utricles and three basal turns of fetal cochleas were used.

Samples were transferred under sterile conditions to the flat caps of PCR tubes holding a volume of 18 µL of medium (Figure 3). The caps were placed in a humidified well chamber to prevent evaporation. The viral construct was added to the medium (1:10 volume) for a total viral dose of 3.5 × 10^12^ gc, an order of magnitude greater than what has been used for non-human primates [13]. Samples were incubated at 37 °C. After 48 h, a full medium change was carried out by removing the caps and culturing the samples in 4 mL of fresh medium added to the six-well plate. On day 4, tissue culture was terminated and samples were fixed overnight in 4% formaldehyde (prepared from paraformaldehyde) in 0.1 M sodium/potassium phosphate buffer (pH 7.4) at 4 °C (Figure 3). Samples were kept at 4 °C and shielded from light until further processing. All samples were decalcified overnight in 10% EDTA.2Na (Sigma-Aldrich, St. Louis, MO, USA) in distilled water (pH 7.4) at 4 °C to dissolve any remaining osseous debris and/or otoconia. 

### 2.3. Immunostaining and Imaging 

Samples were permeabilized for 10 min using 10% DMSO (Thermo Scientific, Rockford, IL, USA) in PBS, followed by blocking for 1 h at room temperature with 5% bovine serum albumin (BSA; Sigma-Aldrich, St. Louis, MO, USA) and 0.05% Tween-20 (Promega, Madison, WI, USA) in PBS. Next, samples were incubated with rabbit anti-myosin 7A (1:350, Novus cat# NBP1-84266; or 1:100, Proteus, cat# 25-6790) for 4 days at room temperature in a humidified chamber. Samples were then washed three times for 15 min and incubated with Alexa Fluor^TM^ 594-conjugated donkey anti-rabbit immunoglobulin for 3 days at room temperature (1:500, Invitrogen cat# A21207). After three wash steps for 15 min, nuclei were stained with 4′,6-diamidino-2-phenylindole (DAPI; 1:1000, Vector Laboratories Ltd., Peterborough, UK) for 15 min at room temperature. After incubation, one short wash and three 15 min wash steps were carried out. Explants were then transferred to a glass slide covered with a raised cover slip and mounted using Prolong Diamond Antifade Mountant (Thermo Scientific, Rockford, IL, USA). This enabled visualization of the surface of the explants without damaging the sample. 

### 2.4. Image Acquisition and Processing

Pseudocolor images of explant tissues were acquired with a Leica SP8 confocal laser scanning microscope using Leica objectives (20×/0.7 dry HC PL Apo, 40×/1.3 oil HC PL Apo CS2, 63×/1.4 oil HC PL Apo or 100×/1.3 oil HC PL Fluotar) and operated under Leica Application Suite X microscope software (LAS X, Leica Microsystems, Buffalo Grove, IL, USA). The laser intensity ranged from 1 to 4% for different channels, with a HyD gain of 10–25. Areas within the sample were randomly selected for acquisition. For cochlear samples, the presumed tonotopical locations of acquisition ranged between 20,000 Hz and 4000 Hz. Maximal projections were obtained from image stacks with optimized z-step size. Brightness and contrast adjustments were performed with Fiji (ImageJ version 1.52p) or Adobe Photoshop CC 2018. 

### 2.5. Quantification of eGFP Expression and Transduction Efficiency 

Cell counts were performed on the separate channels of MYO7A and overlay channels of MYO7A and eGFP, collected from two adult ampullas, three adult utricles, three fetal utricles, and three fetal cochleas. eGFP-positive hair cells were quantified as a percentage of the total number of MYO7A-positive hair cells in all samples. All counts were performed independently by three investigators (WV, TV, and MM) in a blinded fashion. The mean values were taken to plot the data. 

## 3. Results

### 3.1. AAV9-PHP.B Vectors Transduce Cells in Adult human Vestibular Organs 

To investigate whether AAV9-PHP.B vectors were able to transduce human inner ear cells, we first examined their efficacy in adult utricles collected during vestibular schwannoma surgery. The transduction of hair cells was assessed by immunostaining with antibodies directed against the hair cell marker MYO7A. Low-magnification confocal images showed eGFP signal throughout the entire utricular macula, indicating that transduction was efficacious (Figure 4). Next, we repeated cultures and immunostaining with three additional surgical samples. We found that AAV9-PHP.B drives robust eGFP expression in adult utricles and ampullas (Figure 5). Both hair cells and supporting cells showed eGFP expression, albeit to varying degrees, indicating that transduction efficacies may differ between the various cell types. Other collected surgical samples were either too damaged or did not survive cell culture. 

### 3.2. AAV9-PHP.B Vectors Transduce Cells in the Fetal Human Inner Ear 

In order to evaluate AAV9-PHP.B viral transduction in fetal human inner ear specimens, the cochlea and vestibular organs were dissected (Figure 2C–E) and cultured, followed by immunostaining. In line with the results obtained with adult utricles, the fetal inner ear samples showed robust eGFP expression as well (Figure 6). In cochlear samples, the inner and outer hair cells showed less eGFP signal than the supporting cells surrounding the sensory domain, indicating different transduction efficacies between sensory cells and non-sensory cells. Vestibular samples showed strong eGFP expression in both hair cells and supporting cells, in the saccule as well as the utricle (Figure 6). These findings were confirmed in two additional samples of the same fetal age (data not shown). 

### 3.3. Quantification of eGFP-Positive Hair Cells and Transduction Efficiency 

To quantify transduction efficiency, cell counts were performed on 11 samples by three blinded investigators. eGFP-positive hair cells were quantified as a percentage of the total number of remaining MYO7A-positive hair cells in adult utricle and ampulla samples and in fetal utricle and cochlear samples (Figure 7). 

In the adult ampulla, the percentage of transduced hair cells ranged between 54% and 62%. The adult utricle showed a hair cell transduction efficacy ranging between 42% and 80%. In the fetal samples, percentages of eGFP-positive hair cells amounted to 22–52% and 53–69% for cochlear and utricular samples, respectively (Figure 7). Interobserver variability was 8.3–25.3%. 

## 4. Discussion

We developed a novel culture model to investigate the transduction efficacy of AAV9-PHP.B vectors in ex vivo explants of fetal and adult human inner ear sensory epithelia. The culture method allowed for survival of inner ear tissue excised at fetal and adult stages and allowed us to minimize the volume of culture media in which viral vectors were diluted. We found that, on average, approximately one out of two hair cells in adult vestibular samples showed eGFP expression (61%), with a comparable transduction efficacy in utricles (63%) and ampullas (58%). In fetal samples, efficacy was largely similar for utricular hair cells (59%). Efficacy of fetal cochlear hair cells seemed less effective (37%) than vestibular hair cells. Our data suggest that AAV9-PHP.B vectors are promising candidates for targeted gene delivery for the treatment of genetic inner ear disorders in both fetal and adult patients and perhaps at developmental stages in between. Furthermore, the data demonstrate that the constitutively active CMV promoter can efficiently drive the expression of eGFP in human hair cells and supporting cells in vitro. As such, our data are consistent with in vivo data from mouse and non-human primate inner ears, showing that both the AAV9-PHP.B capsid and the CMV promoter are efficacious [13,14,17]. We did note lower hair cell transduction rates in our human in vitro experiments than has been reported for the prior in vivo work [13,14,17]. This may be due to the in vitro experimental paradigm, as other in vitro work with adult human tissue has also shown lower transduction rates compared to those reported in vivo [4,5,6]. 

A difference in the mean total number of hair cells was observed between the fetal and adult utricles counted by investigators. In fetal samples, the mean total number of utricular hair cells ranged from 364 to 614 in processed images, whereas in adult samples, these ranged from 98 to 216. This difference may be explained by several factors. Firstly, the adult utricles underwent degeneration caused by aging, whereas the fetal tissues did not. Secondly, the collection of utricles from adult inner ears was more traumatic than that of fetal inner ears; in adults, the otic capsule consisted of bone that had to be drilled out, whereas the fetal inner ears, which were cartilaginous in this stage, were easily cut open by micro-tweezers. Thirdly, adult samples were collected from patients with a vestibular schwannoma. Studies have shown that there is significant degeneration of type I and II vestibular hair cells and cochlear inner and outer hair cells due to direct and indirect effects of this tumor [18,19]. 

A number of drawbacks arise from working with human inner ear specimens. In order to contribute to reproducibility, these drawbacks need to be addressed. Firstly, when working with cell lines, culture conditions can be optimized with relative ease, whereas when using human inner ear specimens, research depends on collection of sufficient numbers of specimens in order to test a limited number of variables. Our culture protocol for human inner ear explants allowed for the current investigation but could possibly be optimized further. Secondly, since collection of these specimens occurs on a weekly basis, experiments have to be carried out sequentially instead of concurrently. This could increase inter-experimental variability. Lastly, these small and fragile explants lack the protection of an otic capsule and hence are prone to damage. 

Despite these drawbacks, we suggest that human inner ear specimens may be a useful model system for facilitating clinical translation of gene therapy vectors whose efficacy in translating eGFP and/or other gene constructs has already been established in animals or other model systems. The human inner ear culture system may be another tool to complement other more established gene therapy paradigms, including in vivo model systems (such as transgenic mice and non-human primates) and in vitro systems (such as cell lines and inner ear organoids). We suggest that positive data demonstrating viral transduction and expression of exogenous gene constructs may provide compelling evidence supporting the advancement of gene therapy reagents toward clinical application. 

## Figures and Tables

**Figure 1 biomolecules-12-00816-f001:**
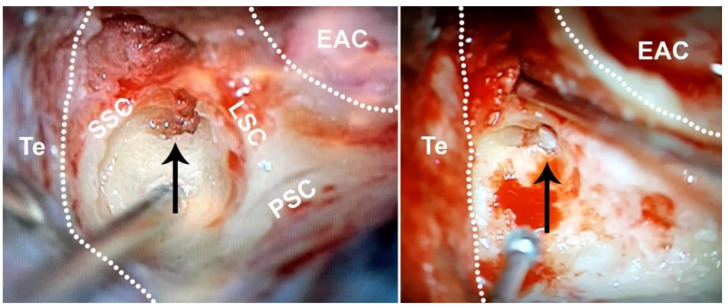
The adult bony labyrinth, right ear. Adult vestibular specimens were collected from the opened semicircular canals (black arrow, left panel) and after further drilling of the vestibulum (black arrow, right panel). EAC, external auditory canal. LSC, lateral semicircular canal. PSC, posterior semicircular canal. SCC, superior semicircular canal. Te, tegmen.

**Figure 2 biomolecules-12-00816-f002:**
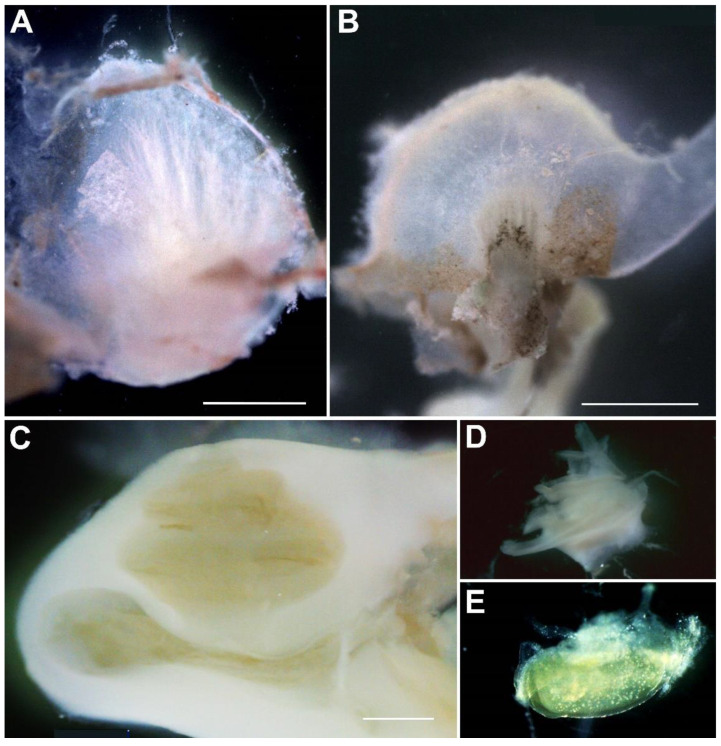
Human adult and fetal dissected inner ears. (**A**) Adult utricle. (**B**) Adult ampulla. (**C**) Fetal cochlea, W14. (**D**) The otic capsule was removed to expose the modiolus and organ of Corti. (**E**) Fetal saccule, W14. Scale bars: 1 mm.

**Figure 3 biomolecules-12-00816-f003:**
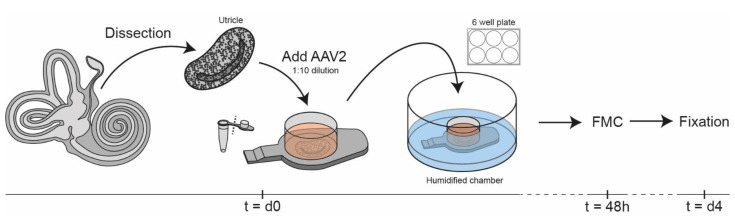
Experimental setup. Tissues were dissected and transferred to a flat cap of a PCR tube. Viral vector was added to the medium in a concentration of 1:10. After 48 h of incubation, 4 mL of medium was added, followed by fixation on day 4. FMC: full media change.

**Figure 4 biomolecules-12-00816-f004:**
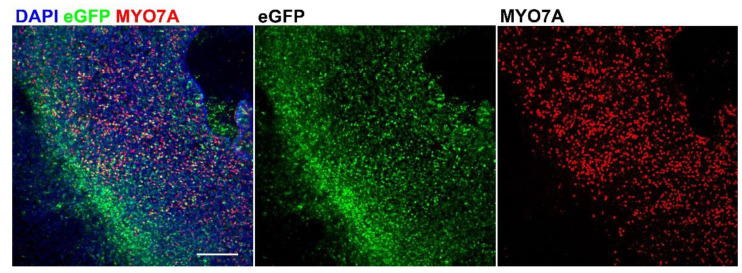
Overview of transduced adult utricle, immunostained with anti-MYO7A antibodies. Transduction was effective, as indicated by eGFP expression throughout the specimen. Scale bar applies to all images: 200 µm.

**Figure 5 biomolecules-12-00816-f005:**
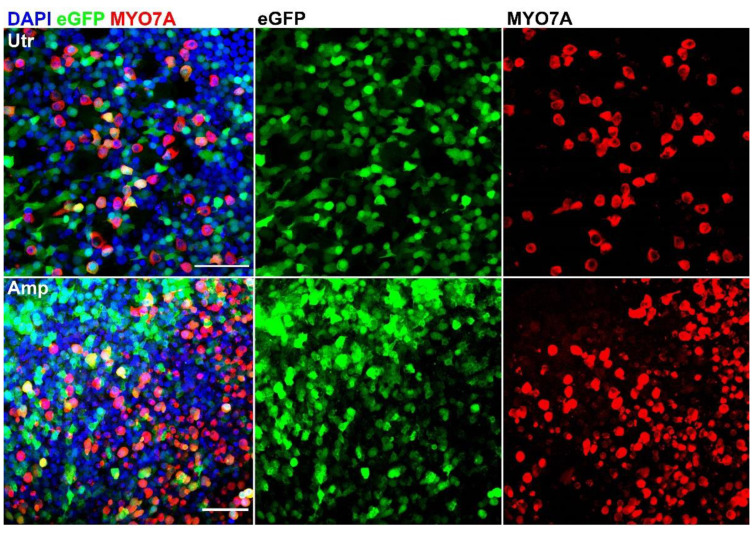
Details of transduced adult utricle (Utr) and ampulla (Amp), immunostained with anti-MYO7A antibodies. Cell counts were carried out on data from 2 ampullas and 3 utricles. Scale bars apply to all images in each row: 50 µm.

**Figure 6 biomolecules-12-00816-f006:**
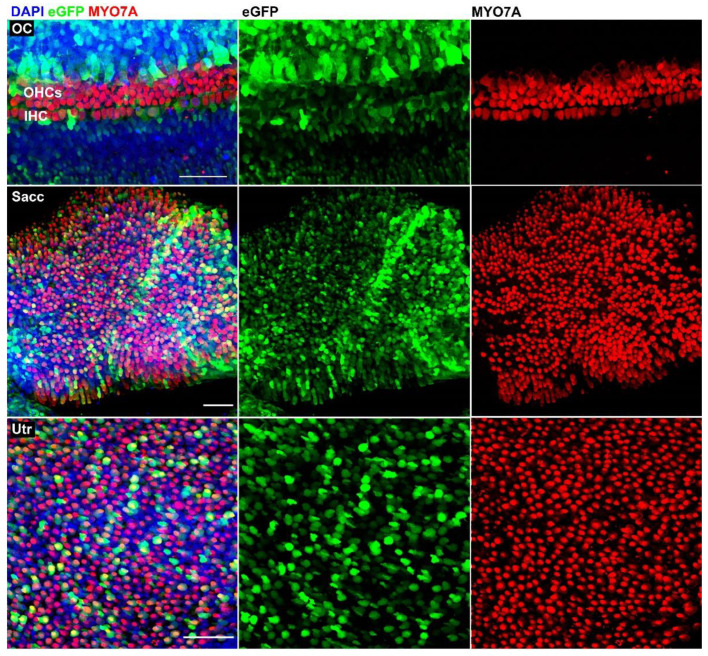
Details of transduced fetal organ of Corti (OC) showing inner and outer hair cells, respectively (IHC, OHC), and a fetal saccule (Sacc) and utricle (Utr), immunostained with anti-MYO7A antibodies. Fetal age = 14 weeks. Scale bars apply to all images in each row: 50 µm.

**Figure 7 biomolecules-12-00816-f007:**
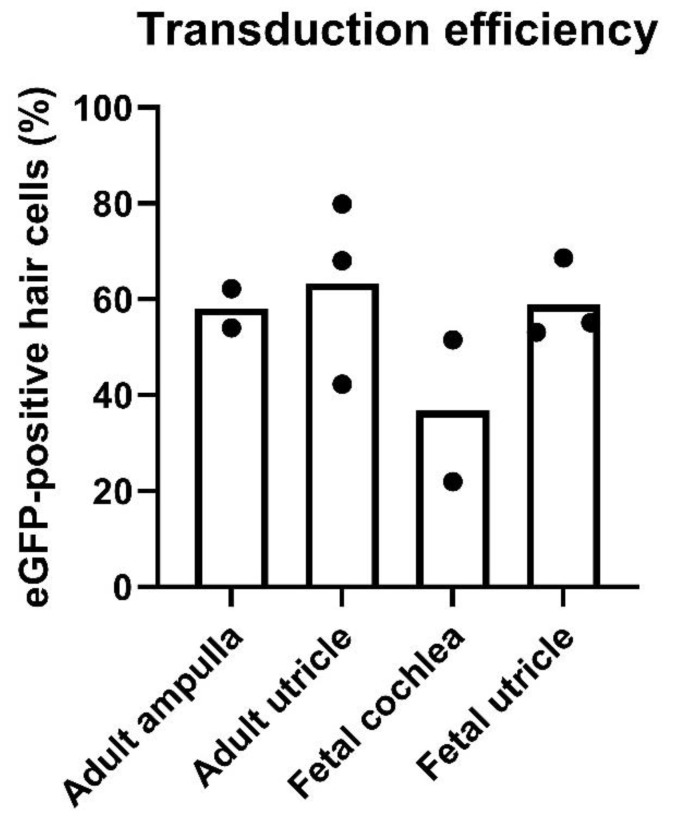
Transduction efficiency. The percentage of eGFP-positive hair cells was quantified by three blinded investigators. Cell counts on adult specimens were carried out on data from two ampullas and three utricles. Cell counts on fetal specimens were carried out on data from two cochleas and three utricles. Each black dot represents the mean of three individual cell counts. Bars represent the mean for each end organ, as indicated on the x-axis. eGFP-positive hair cells (%): adult ampulla, 58%; adult utricle, 63%; fetal cochlea, 37%; fetal utricle, 59%.

## Data Availability

The data presented in this study are available on request from the corresponding author.

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
