# Peer review of "Efficient Viral Transduction in Fetal and Adult Human Inner Ear Explants with AAV9-PHP.B Vectors"

_biomolecules, 2022, doi:10.3390/biom12060816_

Round 1

Reviewer 1 Report

This manuscript draft describes the use of a potent synthetic adeno-associated virus (AAV) serotype, namely GFP-expressing AAV9-PHP.B, in human inner ear tissue. While a plethora of preclinical experimental data to demonstrate the feasibility of AAV-mediated cochlear and vestibular gene therapy exists (primarily in murine animal models, but also in rhesus monkeys), the reported use in human tissue is limited and has primarily focused on slightly more accessible vestibular tissue. Here, the authors developed techniques to extract human fetal and adult inner ear tissue, which subsequently was cultured and exposed to the AAV. The fact that a successful transduction could be observed is important from a translational perspective and underlines how promising these novel therapies are. Thus, in summary, the authors present a convincing translational research paper focusing on the use of AAV9-PHP.B in human tissue. However, the transduction of several AAV serotypes in human adult vestibular tissue has been analyzed before. The points that still should be clarified are highlighted below:

Major:

- Overall, the statements should be toned down a bit. After reading the abstract, I had anticipated to see data on adult cochleae as well, which was not the case.

Minor:

AUTHOR LIST

- There appears to be a superfluous space in Professor Holt’s last name (“Hol t”)?

ABSTRACT

- “...translation to humans has not yet been demonstrated.” At least some efforts (primarily with vestibular tissue) have been made here as the authors outline in the introduction.

- The abstract talks about GFP, but it should be eGFP throughout the manuscript.

INTRODUCTION

- Hearing loss is usually not classified as a neurological disorder. See, e.g., PMID 30879893.

MATERIALS AND METHODS

- Most scholars agree that the plural of these Latin nouns should be cochleae and ampullae (see, e.g., Google Books Ngram Viewer). 

- “…identifying the otolith organs (utricles)…” doesn’t make any sense. I assume it should be “(utricle and saccule)”?

- What is “a small volume of medium”?

- Figure 3 describes a 6-well plate, but this is not mentioned in the text. The 4 mL volumes were then added to the 6-well plate while the samples still were in there/the flat caps of PCR tubes?

- Were the cochleae mounted without any liquid around them? E.g., Vectashield?

- “Areas within the sample were randomly selected for acquisition.” At least for the cochleae, the authors should try to obtain a frequency-specific map to give the reader some information about the presumed tonotopical location.

- Some image acquisition parameters are missing – laser intensity, etc.

RESULTS

- I assume the authors mean the following: “…were either too damaged OR did not survive cell culture.”

- Line 166: The plural of specimen is specimens.

- At least for the cochlear samples, inner and outer hair cells should be quantified separately. How do the authors explain that the reported transduction efficiency from other groups with different serotypes and even their own publication from several years ago appears to be higher than in this manuscript?

- Line 185: Superfluous space after 42%.

- Line 221: This should be reference 15 and 16 (currently not cited), not 13 and 14.

DISCUSSION

- The authors state that a novel culture system has been developed. It should be clarified what the novelty is compared to established protocols that have been used for similar experiments in the past (references 3-5).

FIGURES

- Figure 3: Define “FMC” as “Full Media Change” in the legend.

- In general: If the scale bar can be applied to all images/panels, write it in the legend.

- Figure 6: It is more conventional to have the inner hair cells at the bottom and the outer hair cells at the top of the image.

Reviewer 2 Report

The manuscript by Van Beelen and co-authors analyzes the transduction efficacy of AAV9-PHP.B in human hair cells in adult vestibular sensory epithelia (ampulle and utricle) and human fetal sensory epithelia (vestibular and cochlear).  The authors provide sufficient evidence for the conclusion that AAV9-PHP.B infects hair cells in these sensory epithelia with a 50—60% transduction efficacy. This is measured by expression of GFP from a constitutive promoter. 

Overall the paper is quite limited in its’ scope as the above is the only parameter measured.

I would recommend postponing publication to include additional data to complement these findings. 

I provide some suggestions below.

However, I understand the constraints to perform this type of experiments with primary inner ear tissue, and as a consequence the lack of "expected" controls. In case additional experiments are not feasible, I would recommend 

1) Rewriting the abstract that now hints to the fact that primary explant culture may serve as a translational model to optimize gene therapy. This is not the case, as the authors also indicate in their discussion. More of a tool to verify findings already obtained in nhp or rodents.

2) Modify the introduction and discussion that are now very limited.

I think the paper would benefit of a more in-depth overview (and related references to the growing body of literature) of gene therapy approaches that have been used, for vestibular or hearing restoration strategies. Both in terms of gene replacement or gene overexpression. Also, the authors may want to cover more in depth the AAV serotypes that have been tested so far, with which transduction efficiency and cell type specificity/trophism.

The discussion is a short summary of the results. While testing gene therapy on primary human cells is an important step to facilitate clinical translation, primary tissue is basically only suitable for verification purposes. The discussion could address other approaches to complement or circumvent the use of primary (fetal or surgical) tissue. 

Material and methods is sufficiently clear.  I Only have 1 question.

Has the AAV concentration for these experiments (1:10 of the medium volume with the specific titer indicated ) been chosen based on mouse work/previous experience? 

A reference to previous work or an explanation of the choice would be good.

Suggestions to complement the data set for publications.

The authors could test on one of the samples (adult or fetal, depending on the best availability), even only in duplicate, the transduction of a specific gene and a selective promoter to verify that the protein of choice is expressed in the right cell with the correct sub-cellular localization.

The authors may want to consider the use of co-staining several markers at once in one sample (wholemount), in order to quantify efficiency of transduction/localization in multiple cell types. It may even be possible to embed the SE for sectioning after transduction in order to assess multiple features.

A comparison with un-transduced samples would be great to assess 1) potential toxicity of the transduction 2) background tissue autofluorescence in the 488 channel.  

Reviewer 3 Report

The authors have developed a promising new technique using explants in the perspective of human gene therapy.

However, there are still some points to improve.

1-Line 50: The authors mention 9 references of proofs of concepts. First, reference 14 is not one of the proofs of concepts. Secondly, the authors forgot 11 other proofs of concepts such as OTOF or CABP2 to mention only these. It would be better to cite a general review in addition.

2-Line 119 and line 121: I am really surprised about the incubation time. 4 days and 3 days, a week total compared to the 15 minutes for DAPI and comparing with Lee et al., 2020 (reference 14). With this incubation time, the background is considerable. Everything has to be positive.

3-I think it would be interesting to add a relative comparison between the results obtained here with those obtained in mice (Lee et al., 2020: ref. 14) and also the other studies (ref 12 and 13) in the discussion section. Such as supporting the fact that the Cmv promoter is better than Cba promoter (ref 12), explaining the low rate of HCs transduction in fetal cochlea compared to result obtained in mice.

4-Line 221: These are not the right references. Ref 15 and 16 but not 13 and 14.

5-For Figures 3, 4, and 5, it would be better to have eGFP in green and MYO7A in red.

Round 2

Reviewer 1 Report

Most of my former suggestions have been taken into consideration and properly balanced in the resubmitted version. I don't have any further comments.

Author Response

We thank the reviewer for the time spend on our manuscript and the suggested improvements! This is much appreciated.

This reviewer has no further comments.

Reviewer 2 Report

I have revised the new version of the manuscript. Even though the scope of the manuscript is very limited, I agree with the authors it is  important to report the findings for fast tracking clinical translation.

I only have minor comments for this final revision.

Abstract, line 19-20

I would suggest replacing: “Lack of translational models”… and later “A translational model could help”…

With something that stresses the lack of human specific models.

>>Complex in vivo experiments in animal models such as NHP are translational models. I believe the only additive value here is the human (and even adult) origin of the tissue.

Discussion, line 256 

Our previously established culture conditions for human inner ear explants allows the current investigation but could possibly be optimized further

>>>Is there a reference to these “previously established”, or do the author refer to the conditions reported here??, please revise

Discussion, line 271

As mentioned before I would avoid saying human primary tissue is a “platform” but rather than validation on human samples could facilitate clinical translation.

Author Response

We would like to thank this reviewer for the time spend on improving our manuscript. This is greatly appreciated.

We have changed the text of the manuscript according to the final three comments.